# The Influence of Aerated Irrigation on the Evolution of Dissolved Organic Matter Based on Three-Dimensional Fluorescence Spectrum

**Rang Xiao** [1,†], **Hongjun Lei** [2,*,†], **Yongling Zhang** [1], **Zheyuan Xiao** [2], **Guang Yang** [2], **Hongwei Pan** [2], **Yiran Hou** [2], **Jie Yu** [2], **Keping Sun** [2] and **Yecheng Dong** [2]

1 School of Civil Engineering, Research Institute of Water Resources Protection and Utilization in Hexi Corridor, Hexi University, Zhangye 734000, China
2 School of Water Conservancy, North China University of Water Resources and Electric Power, Zhengzhou 450046, China
* Correspondence: leihongjun@ncwu.edu.cn; Tel.: +86-134-6032-3316
† These authors contributed equally to this work.

**Abstract:** In order to unravel the effect of aerated irrigation on soil dissolved organic matter (DOM) fluorescence characteristics, and humification degree, a randomized block experiment was conducted with three factors and a two-level design, i.e., two irrigation rates (0.6 and 1.0 times of crop evaporation pan coefficient, $W_1$ and $W_2$), two nitrogen application rates (225 and 300 kg hm$^{-2}$, $N_1$ and $N_2$), and two aeration rates (15% and 0% in control treatment, $A_1$ and $A_0$). Fluorescence regional integration (FRI) and correlation analysis methods were used to investigate the evolution characteristics of the soil DOM fluorescence spectrum. Under aerated and conventional subsurface irrigation, soil DOM components were dominated by humic acid-like substances, fulvic acid-like substances, tryptophan-like proteins, and supplemented by tyrosine-like proteins and dissolved microbial metabolites. Soil aeration could promote the consumption of soil DOM components under low irrigation rates and accelerate the consumption of soil DOM components under high irrigation rates. The humification index of AI treatments varied from 8.47 to 9.94 during the maturity growth stage of pepper, averagely increased by 31.59% compared with the non-aeration treatment. To sum up, aerated irrigation can promote the depletion of small molecular proteins and accelerate nutrient turnover and the accumulation of big molecular proteins.

**Keywords:** fluorescence characteristics; humification degree; fluorescence regional integration; subsurface drip irrigation; soil DOM

## 1. Introduction

Dissolved organic matter (DOM) is the most important component of agricultural ecosystems, and its migration and transformation are among the most fundamental processes in the biogeochemical cycle [1,2]. It not only reflects soil quality and productivity, but it also influences the exercise of land ecological function. Furthermore, DOM is an important reference for determining soil quality [3]. It is made up of a variety of substances with varying structures and molecular weights [4]. Although soil DOM accounts for a very small fraction of soil organic matter (SOM) (<5%), it is an extremely active and mobile component of agricultural SOM [5]. DOM is more sensitive to the response of agricultural management measures than SOM, and it can reflect the early potential change of SOM [6]. Because the molecular weight is low, microorganisms can directly utilize the unstable components [7].

Soil water content is closely related to the soil oxygen content in agricultural production. Under conventional subsurface drip irrigation (CSDI), irrigation water expels soil air during the irrigation process or several hours later [8]. Under high soil moisture conditions,



soil hampered the exchange of $O_2$ and $CO_2$ in the soil–root–atmosphere system. Even in CSDI, it was extremely simple to cause hypoxia [9,10]. High crop yield [11,12] is difficult to achieve, limiting the long-term development of facility agriculture. Two of the most important farming practices were irrigation and fertilization [13]. Aerated irrigation (AI) has the advantages of CSDI and has the potential to improve soil enzyme activity and microbial abundance while regulating soil aeration [14]. It was also an effective method for improving the soil microenvironment [15]. The impact of microorganisms on soil organic carbon was contentious. Aerated irrigation has gradually gained popularity as drip irrigation technology has advanced in recent years. Compared to CSDI, AI can improve soil aeration, increase crop water use efficiency, and effectively alleviate hypoxia in the rhizosphere soil [16–18], providing technical support for improving crop yield and water use efficiency. It has emerged as the pinnacle of drip irrigation in facility agriculture. AI can promote the increase in photosynthetic rate, which leads to the increase in the dry matter accumulation and yield of plants, so that crops can make better use of soil nutrients [19].

The analysis of DOM composition and humification degree dynamics using AI was beneficial in understanding the mediation role of microorganisms in agroecosystem soil carbon turnover. Three-dimensional fluorescence spectroscopy (3D-EEM) is a technique for decomposing 3D-EEM into individual fluorescence components, which can aid in substance visualization in the fluorescence intensity state. It is used to analyze components and mechanisms with varying fluorescence properties in soil DOM [20]. Previous studies on soil DOM components primarily focused on soil, compound fertilizer, and other single environmental media [21]. Such use of compound fertilizer could improve soil humification degree and the level of soil nutrient supply. Fertilization resulted in terrestrially derived DOM in high-yield fields, bio-sourced DOM in low-yield fields, and reduced DOM humification in medium-low-yield fields [22]. Besides, the ratio of fulvic acid-like and humic acid-like fluorescence peak intensities in water-soluble SOM could characterize its structural changes, with a larger ratio indicating a greater degree of aging and the humification of humic substances in water-soluble SOM [23]. Previous research on soil DOM components primarily focused on soil, compound fertilizer, and other single environmental media. In contrast, research on the effect of AI on farmland primarily focused on soil aeration improvement, soil moisture, nutrient regulation, and physiological effect on crops, among other things. Fluorescence spectroscopy has yet to be used to investigate the effect of AI on the evolution and structural changes of soil DOM fluorescent components.

The composition and phase of DOM were measured using 3D-EEM in this study to investigate the DOM composition and structure of the greenhouse pepper root zone under aerated irrigation. We hypothesized that AI would promote the transformation of small molecular substances to macromolecular substances and thus increase the degree of DOM humification.

## 2. Materials and Methods

### 2.1. Experimental Site

The field experiment was conducted on pepper in a modern greenhouse from 1 April 2021 to 18 July 2021 at the Agricultural High-Efficiency Water Use Test Site of the North China University of Water Resources and Hydropower, Zhengzhou City, Henan Province (34.47′5.91″ N, 113.47′20.15″ E). The solar greenhouse with a total area of 537.6 $m^2$ is a ridge-type structure that spans 9.6 m and has a bay of 4 m. As is common for the area, the greenhouse is configured east–west to trap maximum solar radiation. Fans were installed in the south and damp curtains were installed in the north to regulate the indoor temperature and air humidity. Temperature and humidity changes during the growth period of hot pepper were shown in Figure 1.

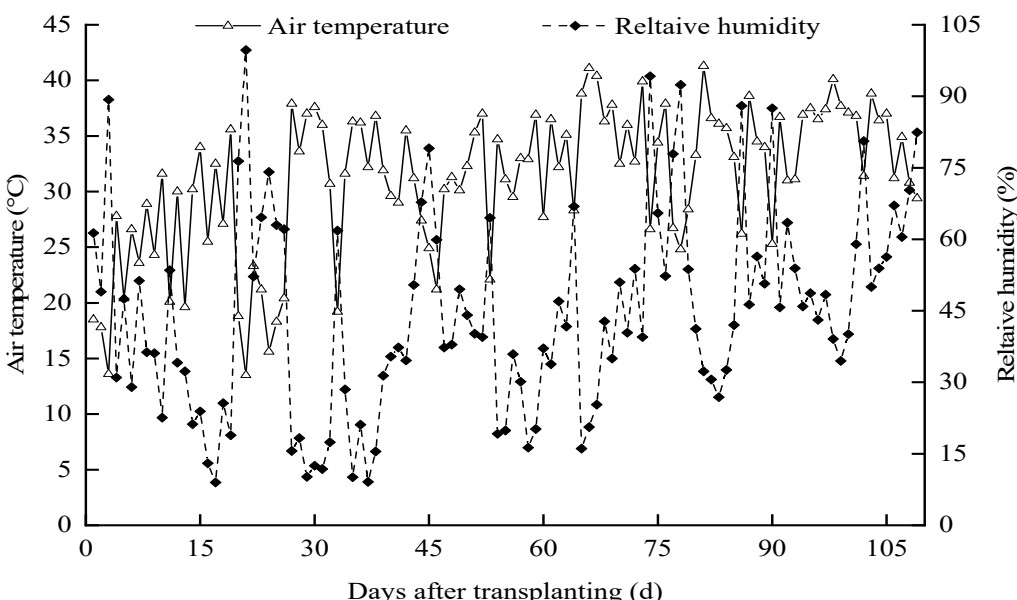

**Figure 1.** Dynamics of the air relative humidity and temperature during the cropping season.

### 2.2. Test Material

The mass fractions of soil sand (0.02–2 mm), silt (0.002–0.02 mm), and clay (<0.002 mm) used in the experiment were 32.99%, 34.03%, and 32.98%, respectively. The soil pH was determined using a glass electrode in a 2.5:1 water: soil suspension. SOM was measured by using $K_2Cr_2O_7$ oxidation–reduction titration. Total soil nitrogen was determined using the Kjeldahl method. Total phosphorus in soil was determined by using molybdenum antimony inverse colorimetry. Total potassium in soil was determined by using flame photometry. Soil bulk density and field capacity was measured using the circular cutting method. The measured soil material content is shown in Table 1.

**Table 1.** The basic physical and chemical properties in the 0–40 cm soil layer.

| Soil Layer/cm | pH | Soil Bulk Density/g·kg$^{-1}$ | SOM/g·kg$^{-1}$ | TN/g·kg$^{-1}$ | TP/g·kg$^{-1}$ | TK/g·kg$^{-1}$ | Field Capacity/% |
|---|---|---|---|---|---|---|---|
| 0–20 | 7.75 | 1.43 | 20,14 | 1.04 | 0.93 | 28.69 | 27.6 |
| 20–40 | 7.82 | 1.49 | 19.25 | 1.12 | 1.22 | 32.24 | 28.4 |

### 2.3. Experimental Design

A three-factor and two-level experiment was set up in a randomized split-plot arrangement, with a total of eight treatments and three replications. The aerated and conventional subsurface drip irrigation were randomly assigned to the main plots. Inorganic nitrogen application rate is 225 kg hm$^{-2}$ for $N_1$ treatment and 300 kg hm$^{-2}$ for $N_2$ treatment. According to the irrigation amount between two irrigation intervals, the evaporating pan coefficients 0.6 and 1.0 are set as $W_1$ and $W_2$ (Table 2), respectively.

A total of 24 plots were included in the experiment. Each plot is 4 m in length, 0.6 m in width, and 2.4 m$^2$ in area. The row spacing was 60 cm and plant spacing was 33 cm. Twelve plants were planted in each plot. CSDI was used for water supply in the plot. The buried depth of drip irrigation belt was 15 cm [24], the rated flow rate of drip head was 1.2 L h$^{-1}$. The dripper spacing was 33 cm, and the rated working pressure was 0.10 MPa. Crops were planted 10 cm from the water dripper parallel to the drip irrigation belt.

**Table 2.** Experimental design.

| Treatment | Irrigation Amount W/mm | Aerated Irrigation Air Void Fraction/% | Nitrogen N/kg hm$^{-2}$ |
|---|---|---|---|
| $W_1A_0N_1$ | $0.6\,E_p$ | 0 | 225 |
| $W_1A_1N_1$ | $0.6\,E_p$ | 15 | 225 |
| $W_2A_0N_1$ | $1.0\,E_p$ | 0 | 225 |
| $W_2A_1N_1$ | $1.0\,E_p$ | 15 | 225 |
| $W_1A_0N_2$ | $0.6\,E_p$ | 0 | 300 |
| $W_1A_1N_2$ | $0.6\,E_p$ | 15 | 300 |
| $W_2A_0N_2$ | $1.0\,E_p$ | 0 | 300 |
| $W_2A_1N_2$ | $1.0\,E_p$ | 15 | 300 |

Note: $W_1$ and $W_2$ were low and high irrigation water treatment (0.6 and 1.0- times crop evaporation pan coefficient), respectively; $A_0$ and $A_1$ were conventional and aerated subsurface drip irrigation treatments; $N_1$ and $N_2$ were low and normal nitrogen application rates (225 and 300 kg hm$^{-2}$), respectively.

### 2.4. Irrigation Method

AI used aerated water by CSDI system (Figure 2), including storage lines, circulation pumps, Mazzei air injector 684 (Mazzei Injector Company, Bakersfield, CA, USA), and other equipment to circulate aeration for 20 min to produce irrigation water with an air void fraction of 15% [25], while CSDI used groundwater for irrigation with an independent water supply device. Irrigation occurred once every 7–10 days based on cumulative evaporation between two irrigation intervals in evaporation dish (E-601 evaporator, 61.8 cm in diameter and 69.0 cm in height). Each plot was equipped with a flow meter and control valve to control the water volume. The working pressure of the irrigation process was 0.10 MPa, and the irrigation amount was measured with a drip meter.

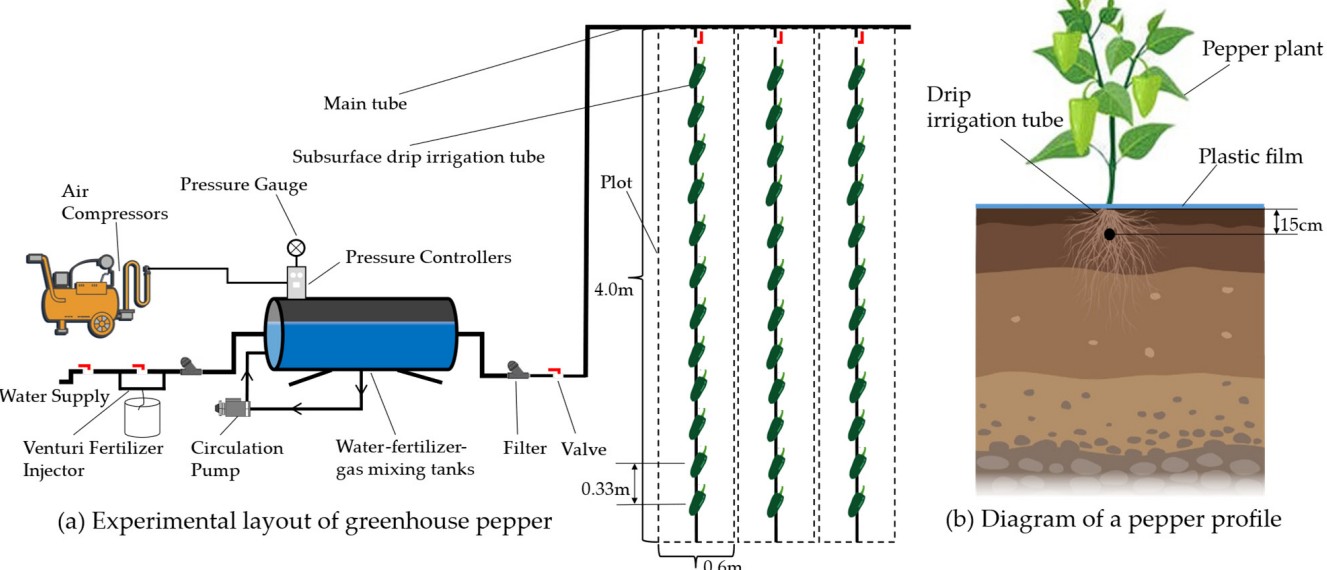

(a) Experimental layout of greenhouse pepper  (b) Diagram of a pepper profile

**Figure 2.** Diagram of experimental layout.

### 2.5. Trial Management

The tested pepper (*Capsicum annuum* L.) variety was "Liangyu". The perlite matrix was used for seedlings cultivation from the commercial producer. The pepper seedlings were transplanted at the age of 4–leaf or 5–leaf and 1 heart. Soil was watered well on the day of transplantation, and the film was covered 13 days after transplanting. The growth period of pepper covered 109 d, and the division of growth period was listed in Table 3.

**Table 3.** Pepper growth period.

| Growth Period | Start Time | End Time | Days after Transplanting/d |
|---|---|---|---|
| Seedling stage | 1 April 2021 | 25 April 2021 | 1–25 |
| Flowering and fruiting stage | 26 April 2021 | 20 May 2021 | 26–50 |
| Fruit expansion stage | 21 May 2021 | 20 June 2021 | 51–81 |
| Maturity stage | 21 June 2021 | 18 July 2021 | 82–109 |

Fertilizers used in the test were urea (N $\geq$ 46%), calcium superphosphate (P$_2$O$_5 \geq$ 12%), and total water-soluble potassium sulfate (K$_2$O $\geq$ 52%). Two nitrogen rates were used, 225 kg hm$^{-2}$ for N$_1$, 300 kg hm$^{-2}$ for N$_2$ treatment, and one rate of phosphate fertilizer of 85 kg hm$^{-2}$ and potassium fertilizer 150 kg hm$^{-2}$ were applied to all treatments. The potassium fertilizer and calcium superphosphate were applied at the beginning as basal, while the N split rate were applied in the irrigation water solution on the 31st, 42nd, 57th, 62nd, 69th, 81st, and 96th day g according to the ratio of 2:3:2:2:3:2:1.

The irrigation amount was calculated according to Equation (1) [26]:

$$W = A \times E_P \times K_P \tag{1}$$

where $W$ is the irrigation amount during two irrigation events, L. $A$ is the planting area of the plot, m$^2$. $E_P$ is the cumulative evaporation from the E-601 evaporation dish within one irrigation cycle, mm. $K_P$ was the coefficient of evaporating dish, 0.6 for W$_1$ treatment and 1.0 for W$_2$ treatment.

*2.6. Test Sampling and Measurement Methods*

2.6.1. Soil Sample Collection

During the flowering and fruit-setting period (20 May 2021), the fruit-expanding period (20 June 2021), and the maturing period (18 July 2021), the root systems of five plants with uniform growth were excavated in each treatment plot. Then the soil attached to the roots was shook down vigorously and the plant residues removed and fully mixed into sealed bags and taken back to the laboratory for air drying.

2.6.2. Extraction of Soil DOM

Air-dried soil samples were passed through a 2 mm sieve and 5 g soil sample were put into a conical bottle. Fifty mL of grade I pure water was added at a water–soil ratio of 5:1 and then shook well and oscillated for 24 h at a constant temperature of 20 °C and 180 r min$^{-1}$. It was then centrifuged for 15 min at 4000 r min$^{-1}$. After centrifugation, the supernatant was passed through a 0.45 μm fiberglass membrane, and the filtrate was the soil DOM solution.

The 3D-EEM of soil DOM was determined using a fluorescence spectrophotometer (Hitachi F-4600) for spectral scanning. The parameters were set as follows: excitation wavelength Ex: 200–450 nm, emission wavelength Em: 280–550 nm, scanning speed: 12,000 nm min$^{-1}$, and scanning intervals at 5 nm. Pure water was used as blank.

2.6.3. Fluorescence Spectroscopy Analysis and Data Processing

Data Processing and Volume Integration

The scanned three-dimensional fluorescence spectra data were blank with pure water, and the measured fluorescence data were pre-processed with MATR2009a (MathWorks, Natick, MA, USA) to eliminate Rayleigh and Raman scattering. In order to quantitatively analyze the content of each component of soil DOM in each growth period, the three-dimensional fluorescence spectrogram of soil DOM was divided into five characteristic regions (Table 4) [27].

$$\phi_i = (\lambda_{Ex}/\lambda_{Em})d\lambda_{Ex}d\lambda_{Em} \tag{2}$$

$$\phi_{i,n} = \phi_i \cdot MF_i \tag{3}$$

where $i$ = I, II, III, IV, V, where I ($\lambda_{Ex}/\lambda_{Em}$) is the fluorescence intensity at excitation wavelength $\lambda_{Ex}$ and emission wavelength $\lambda_{Em}$, $d\lambda_{Ex}$ and $d\lambda_{Em}$ were the data intervals of excitation and emission wavelengths in the fluorescence matrix, respectively. The volume $\phi_i$ was multiplied with $MF_i$ to obtain the normalized volume $\phi_i,n$, where $MF_i$ is equal to the inverse of the area of each region of the excitation–emission matrix as a percentage of the total region.

**Table 4.** Characteristics of fluorescent peak regions in three-dimensional fluorescence spectra of dissolved organic matter.

| Fluorescent Area | $\lambda_{Ex}/\lambda_{Em}$/nm | Fluorescent Components |
|:---:|:---:|:---:|
| I | 200–250/280–330 | Tyrosine-like proteins |
| II | 200–250/330–380 | Tryptophan-like proteins |
| III | 200–250/380–550 | Fulvic acid |
| IV | >250/280–380 | Soluble microbial metabolites |
| V | >250/380–550 | Humic acid |

According to Equation (2) [28], the fluorescence spectrogram data in selected fluorescence regions were integrated using the Origin Pro 2022 software, and the volume of specific fluorescence region of soil DOM was calculated ($\phi_i$). According to Equation (3) [28], standardized processing was carried out.

Spectral Index Calculation

(1)    Fluorescence index (*FI*) is calculated [29] as below:

$$FI = \frac{f_{\lambda_{Em}=450nm}}{f_{\lambda_{Em}=500nm}} \lambda_{Ex} = 370nm \tag{4}$$

where $f_{\lambda Em}$ = 450 nm is the fluorescence intensity at excitation wavelength of 370 nm and emission wavelength of 450 nm and $f_{\lambda Em}$ = 500 nm is the fluorescence intensity at excitation wavelength of 370 nm and emission wavelength of 500 nm.

(2)    The formula of autogenous index/biological index (*BIX*) [29] is listed as below:

$$BIX = \frac{f_{\lambda_{Em}=380nm}}{f_{\lambda_{Em}=430nm}} \quad \lambda_{Ex} = 310nm \tag{5}$$

where $f_{\lambda Em}$ = 380 nm is the fluorescence intensity at excitation wavelength of 310 nm and emission wavelength of 380nm and $f_{\lambda Em}$ = 430 nm is the fluorescence intensity at excitation wavelength of 310 nm and emission wavelength of 430 nm.

(3)    Humification index (*HIX*) is calculated [29] as below.

$$HIX = \frac{\sum_{435nm}^{480nm} f_{\lambda_{Em}}}{\sum_{300nm}^{345nm} f_{\lambda_{Em}}} \quad \lambda_{Ex} = 254nm \tag{6}$$

where is the integral value of excitation wavelength of 254 nm and emission wavelength in the range of 435 to 480 nm and is the integral value of excitation wavelength of 254 nm and emission wavelength in the range of 300 to 345 nm.

*2.7. Statistical Analysis*

Microsoft Excel 2019 was used for data analysis and Origin Pro 2022 (Origin Lab Corp. Redmond, MA, USA) was used for plotting. The SPSS 25.0 software (SPSS Inc., Chicago, IL, USA) was used for ANOVA, Duncan's multiple comparisons, and Pearson's correlation analysis with the statistical significance test at $p < 0.05$ and $p < 0.01$.

## 3. Results and Analysis

### 3.1. Effects of Aerated Irrigation on DOM Fluorescence Components in Soil

The three-dimensional fluorescence region integration (FRI) of the DOM of pepper rhizosphere soil in the greenhouse is shown in Figure 3. With the growth of pepper, the relative contents of five major fluorescent components increased first, then decreased, and peaked at the fruit expansion stage. The amount of irrigation had a significant impact on tyrosine-like proteins at the flowering and fruit-setting stage; fulvic-acid-like components at the maturity stage ($p < 0.05$) (Figure 3a, c); tryptophan-like protein components at the flowering and fruit-setting stage, fruit expansion stage, and maturity stage (Figure 3b); tyrosine-like protein components at the fruit expansion stage; and soluble microbial metabolites and humus acid-like components at the mature stage ($p < 0.01$) (Figure 3a,d,e). When compared with $W_1A_0N_1$ and $W_1A_0N_2$, the relative contents of tyrosine-like protein components in soil DOM averagely increased by 43.83% and 13.84% $W_2A_0N_1$ and $W_2A_0N_2$, respectively, and tryptophan-like protein components in soil DOM averagely increased by 35.56% and 16.85% in $W_2A_0N_1$ and $W_2A_0N_2$, respectively, during the whole growth period. In comparison with $W_1A_0N_1$ and $W_1A_0N_2$, the relative contents of fulvic acid components decreased by 19.22%, soluble microbial metabolites decreased by 16.51% and humus-like acid components decreased by 21.68% on average in the soil treated with $W_2A_0N_1$ and fulvic acid components decreased by 8.55%, soluble microbial metabolites decreased by 5.53%, humus-like acid components decreased by13.61% on average in the soil treated with $W_2A_0N_2$ during the growth period. When compared with treatment $W_1A_1N_1$, on average, the relative content of tyrosine-like proteins increased by 34.43%, tryptophan-like proteins increased by 13.27%, fulvic acid-like proteins increased by 15.87%, and humus acid-like proteins increased by 8.06% on average in the soil treated with $W_2A_1N_1$ during the whole growth period. The relative content of soluble microbial metabolites increased by 4.48% in the flowering and fruit-setting stages and decreased by 2.78% in the fruit expansion and maturity stages.

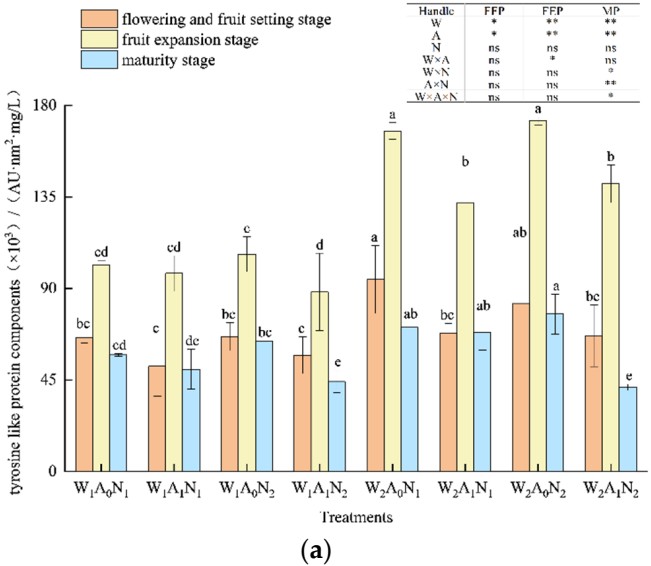

(a)

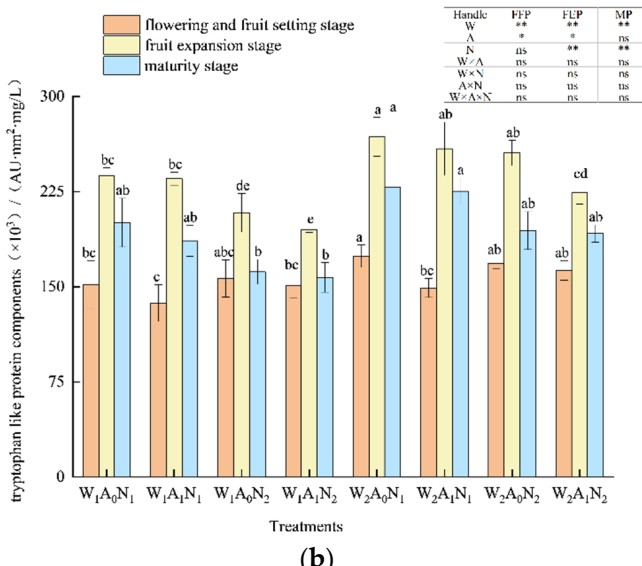

(b)

**Figure 3.** *Cont.*

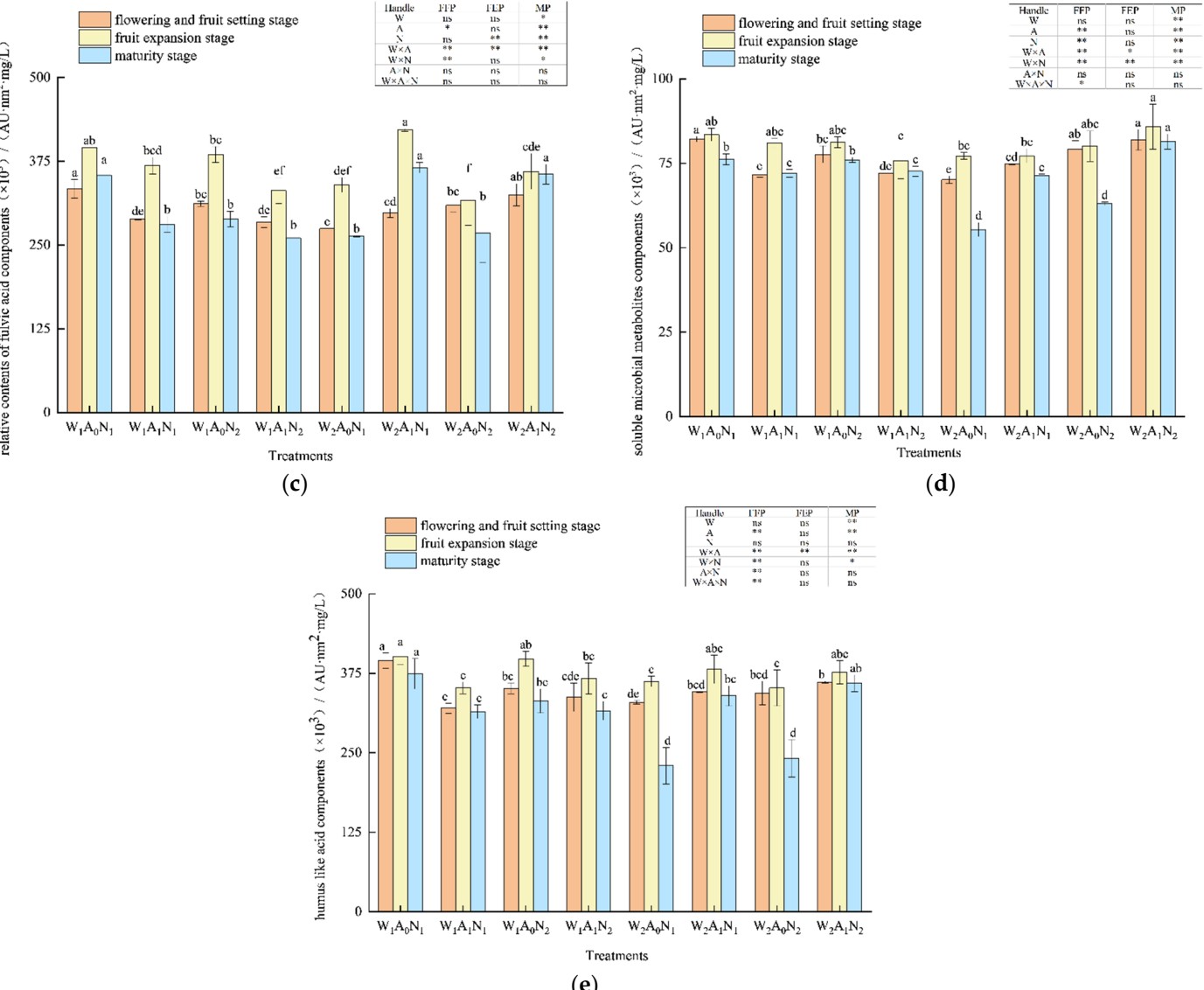

**Figure 3.** Regional integral analysis of soil DOM based on three-dimensional fluorescence spectra Note: $W_1$ and $W_2$ were low and high irrigation water treatment (0.6 and 1.0- times crop evaporation pan coefficient), respectively; $A_0$ and $A_1$ were non-aeration and aerated irrigation treatments; $N_1$ and $N_2$ were low and normal nitrogen application rates (225 and 300 kg $hm^{-2}$), respectively. * and ** identify significant differences at $p < 0.05$ and $p < 0.01$, respectively, and ns identifies no significant difference. (**a**) The amount of irrigation had a significant impact on tyrosine-like proteins at the flowering and fruit-setting stage; (**b**) The tryptophan-like protein components at the flowering and fruit-setting stage, fruit expansion stage, and maturity stage (**c**) The amount of irrigation had a significant impact on fruit-setting stage; (**d**) The tyrosine-like protein components at the soluble microbial metabolites; (**e**) The tyrosine-like protein components at humus acid-like components at the mature stage. Same letters within each column indicate non-significant difference among the treatments.

During the whole growth period, the relative contents of tyrosine-like protein components averagely increased by 23.53%, tryptophan-like protein components increased by 14.94%, fulvic acid-like components increased by 19.90%, soluble microbial metabolite increased by 12.99%, and humic acid-like components increased by 7.77% in the soil treated with $W_2A_1N_2$, compared with treatment $W_1A_1N_2$. During fruit expansion and maturity, flowering and fruit-setting stages, the amount of nitrogen application had an extremely

significant effect on tryptophan-like protein components, fulvic acid-like components, and soluble microbial metabolites ($p < 0.01$) (Figure 3).

In the fruit expansion stage and mature stage, the relative content of tryptophan-like protein component decreased by 14.02% and fulvic acid-like component decreased by 7.61% in soil under normal nitrogen treatment on average, when compared with low nitrogen treatment. When compared with treatment $W_1A_0N_1$, the relative content of soluble microbial metabolites in soil under treatment $W_1A_0N_2$ decreased by 2.88% on average during the flowering and fruit-setting stage and mature stage, compared with treatment $W_1A_1N_1$. Treatment $W_1A_1N_2$ increased the soil soluble microbial metabolites by 0.81% on average in the flowering and fruit-setting stage and maturity stage. At the flowering and fruit-setting stage and maturity stage, the relative content of soil soluble microbial metabolites of treatment $W_2A_0N_2$ and treatment $W_2A_1N_2$, respectively, increased by 13.33% and 11.83%, when compared with treatment $W_2A_0N_1$ and treatment $W_2A_1N_1$. The tyrosine-like protein components, fulvic acid-like components, soluble microbial metabolites, and humic acid-like components were highly affected by soil aeration at fruit expansion and maturity ($p < 0.01$) (Figure 3). The tyrosine-like protein components, tryptophan-like protein components, fulvic acid-like components, and tryptophan-like protein components at the fruit expansion stage had significant effects ($p < 0.05$) (Figure 3) as well. During the whole growth period, the relative contents of tyrosine-like protein components averagely decreased by 12.66%, tryptophan-like protein components averagely decreased by 5.93%, fulvic acid-like components decreased by 13.71%, soluble microbial metabolites decreased by 7.13%, and humic acid-like components decreased by 15.76% in the soil treated with $W_1A_1N_1$ and the relative contents of tyrosine-like protein components decreased by 20.67%, tryptophan-like protein components decreased by 4.21%, fulvic acid-like components decreased by 10.88%, soluble microbial metabolites decreased by 6.08%, and humic acid-like components decreased by 5.47% in the soil with the treatment of $W_1A_0N_2$ when compared with $W_1A_0N_1$ and $W_1A_0N_2$. When compared with treatment $W_2A_0N_1$ and treatment $W_2A_0N_2$, the relative contents of tyrosine-like protein components, respectively, decreased by 17.45% and 6.41% in the soils with treatments of $W_2A_1N_1$ and $W_2A_1N_2$ and tryptophan-like protein components, respectively, decreased by 27.85% and 5.64% in the soils with treatments of $W_2A_1N_1$ and $W_2A_1N_2$. During the growth period, when compared with treatment $W_2A_0N_1$ and treatment $W_2A_0N_2$, the relative contents of fulvic acid components increased by 23.79%, soluble microbial metabolites increased by 11.85%, and humic acid components increased by 19.33% in the soils with treatments of $W_2A_1N_1$, and the relative contents of fulvic acid components increased by 17.15%, soluble microbial metabolites increased by 13.31%, and humic acid components increased by 20.27% in the soil with treatments of $W_2A_1N_2$.

In comparison to the AI treatment, the CSDI treatment revealed the presence of fulvic acid-like substances $C_3$ fraction [30]. The $C_3$ component (two peaks at 280/500 nm and 375/500 nm) is a typical humus-like component of terrestrial origin, with a minor contribution from allochthonous macrophytes. Although the relative molecular weight of the $C_3$ component is greater than that of the $C_1$ and $C_2$ components, the degree of aromatization is lower.

### 3.2. Effects of Aerated Irrigation on DOM Spectral Index in Soil

The fluorescence index (FI), biological index (BIX), and humification index (HIX) were used to assess the impact of AI on the source of soil DOM and humification degree. According to Table 5, the fluctuation range of the DOM fluorescence index under the CSDI and AI during the entire growth period was 1.63–1.76 and 1.66–1.81, respectively; the fluctuation range of the biological index was 0.67–0.72 and 0.65–0.72, respectively, and the fluctuation range of humification index was 5.39–13.69 and 8.47–12.85, respectively, indicating a trend of gradual decrease with the growth process. The humification index of $W_2A_1N_1$ and $W_2A_1N_2$ was 57.14% and 78.46% higher, respectively, than that of $W_2A_0N_1$ and $W_2A_0N_2$ ($p < 0.05$). The humification index of $W_2A_0N_1$ and $W_2A_0N_2$ decreased by

49.58% and 40.62% compared with that of treatment $W_1A_0N_1$ and $W_1A_0N_2$, respectively ($p < 0.05$). HIX in AI varied from 8.47 to 9.94. Averagely, HIX in AI increased by 31.59% compared with that in CSDI.

**Table 5.** Fluorescence spectrum indices of soil DOM at different growth period.

| Irrigation Treatment | | Fluorescence Index (FI) | | | Biological Index (BIX) | | | Humification Index (HIX) | | |
|---|---|---|---|---|---|---|---|---|---|---|
| | | FFS | FES | MS | FFS | FES | MS | FFS | FES | MS |
| CSDI | $W_1A_0N_1$ | 1.70 ± 0.02 a | 1.72 ± 0.06 a | 1.71 ± 0.03 bc | 0.72 ± 0.01 a | 0.69 ± 0.02 ab | 0.71 ± 0.02 ab | 13.69 ± 1.62 a | 12.39 ± 0.96 a | 10.69 ± 1.97 a |
| | $W_1A_0N_2$ | 1.72 ± 0.05 a | 1.69 ± 0.02 a | 1.74 ± 0.05 ab | 0.72 ± 0.01 a | 0.70 ± 0.03 a | 0.72 ± 0.02 a | 11.77 ± 1.77 ab | 11.73 ± 0.54 ab | 9.38 ± 0.48 a |
| | $W_2A_0N_1$ | 1.72 ± 0.04 a | 1.76 ± 0.04 a | 1.63 ± 0.10 c | 0.70 ± 0.01 b | 0.67 ± 0.02 ab | 0.71 ± 0.02 abc | 11.90 ± 1.92 ab | 9.15 ± 0.57 c | 5.39 ± 1.99 b |
| | $W_2A_0N_2$ | 1.70 ± 0.03 a | 1.70 ± 0.03 a | 1.64 ± 0.06 c | 0.71 ± 0.01 ab | 0.67 ± 0.03 ab | 0.69 ± 0.02 bcd | 11.20 ± 1.06 ab | 9.84 ± 1.98 bc | 5.57 ± 2.84 b |
| AI | $W_1A_1N_1$ | 1.72 ± 0.02 a | 1.73 ± 0.04 a | 1.75 ± 0.03 ab | 0.72 ± 0.01 a | 0.68 ± 0.03 ab | 0.72 ± 0.01 a | 11.52 ± 1.99 ab | 10.10 ± 1.27 bc | 9.53 ± 1.48 a |
| | $W_1A_1N_2$ | 1.70 ± 0.06 a | 1.72 ± 0.07 a | 1.74 ± 0.04 ab | 0.72 ± 0.02 a | 0.68 ± 0.03 ab | 0.72 ± 0.01 a | 12.85 ± 1.73 ab | 9.23 ± 1.98 c | 9.53 ± 0.72 a |
| | $W_2A_1N_1$ | 1.70 ± 0.03 a | 1.73 ± 0.09 a | 1.67 ± 0.06 bc | 0.70 ± 0.01 b | 0.66 ± 0.03 ab | 0.67 ± 0.03 d | 12.62 ± 0.41 ab | 10.37 ± 1.82 abc | 8.47 ± 1.53 a |
| | $W_2A_1N_2$ | 1.66 ± 0.03 a | 1.72 ± 0.07 a | 1.81 ± 0.04 a | 0.70 ± 0.01 b | 0.65 ± 0.03 b | 0.68 ± 0.01 cd | 10.37 ± 2.37 b | 9.11 ± 1.48 c | 9.94 ± 1.43 a |

Note: $W_1$ and $W_2$ were low and high irrigation water treatment (0.6 and 1.0-times crop evaporation pan coefficient), respectively; $A_0$ and $A_1$ were conventional and aerated subsurface drip irrigation treatments; $N_1$ and $N_2$ were low and normal nitrogen application rates (225 and 300 kg hm$^{-2}$), respectively. FFS means the flowering and fruiting period; FES means fruit expansion period; MS means mature stage. Data are all expressed as mean ± standard error. Different lowercase letters after the data in the same column indicate a significant difference at $p < 0.05$ level.

### 3.3. Evolution Characteristics of DOM Fluorescence Components in Soil

PARAFAC combined with 2D-COSDOM can obtain the change rule of fluorescence components with time. The load of components of DOM can be obtained using the PARAFAC method (Figures 4 and 5), the synchronous and asynchronous spectrograms of DOM fluorescence components with time were obtained by using the 2D-COS method (Figure 6).

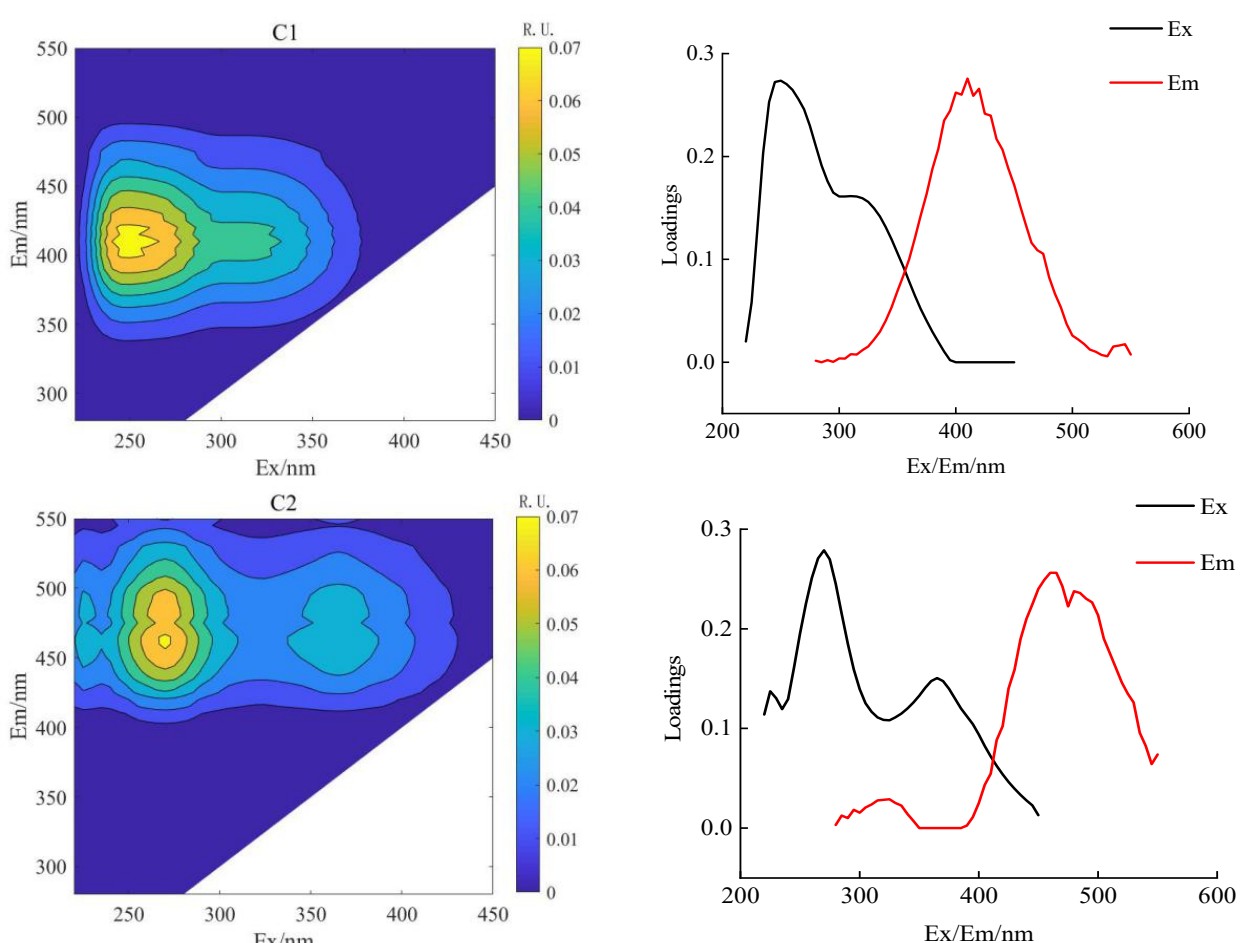

**Figure 4.** Parallel factor analysis of soil DOM in conventional subsurface drip irrigation.

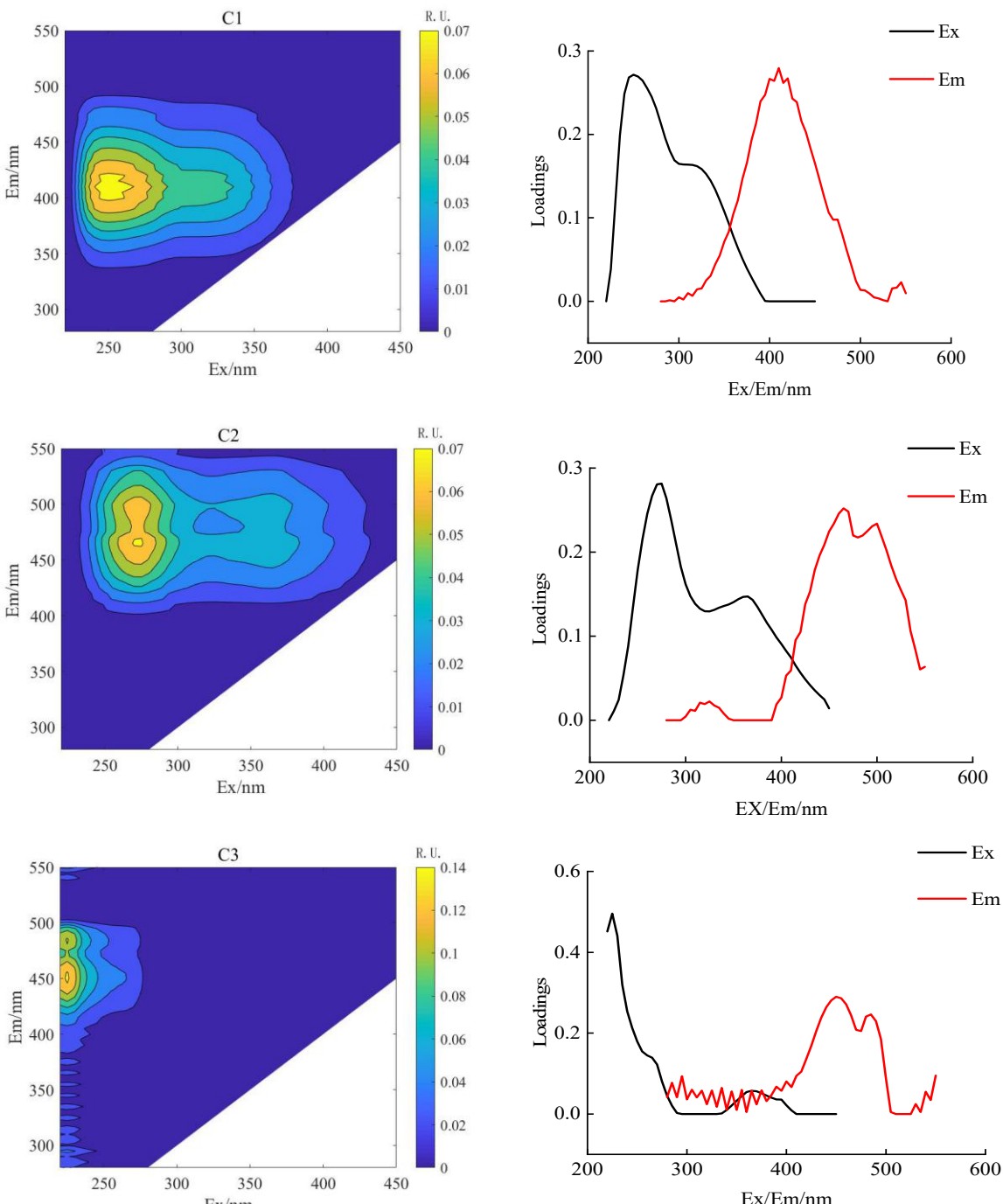

**Figure 5.** Analysis of soil DOM parallel factor in aerated drip irrigation.

According to the spectral reading rules proposed by Noda et al. [31], the changing order of DOM fluorescence components during crop growth was determined (Tables 6 and 7). The fluorescence fractions resolved by PARAFAC were uploaded to the OpenFluor database for comparison with the previous literature. It can be seen that C1 and C2 components both exist in AI and CSDI treatments. The C1 component (325/410 nm) was identified as UV humic-like, a combination of fluorescence peaks A and M under both treatments, probably composed of humic-like compounds derived from biological or microbial activity [32], and the C2 component (two peaks are 260/435 nm and 350/435 nm) identified as terrestrial-derived humic substances [27].

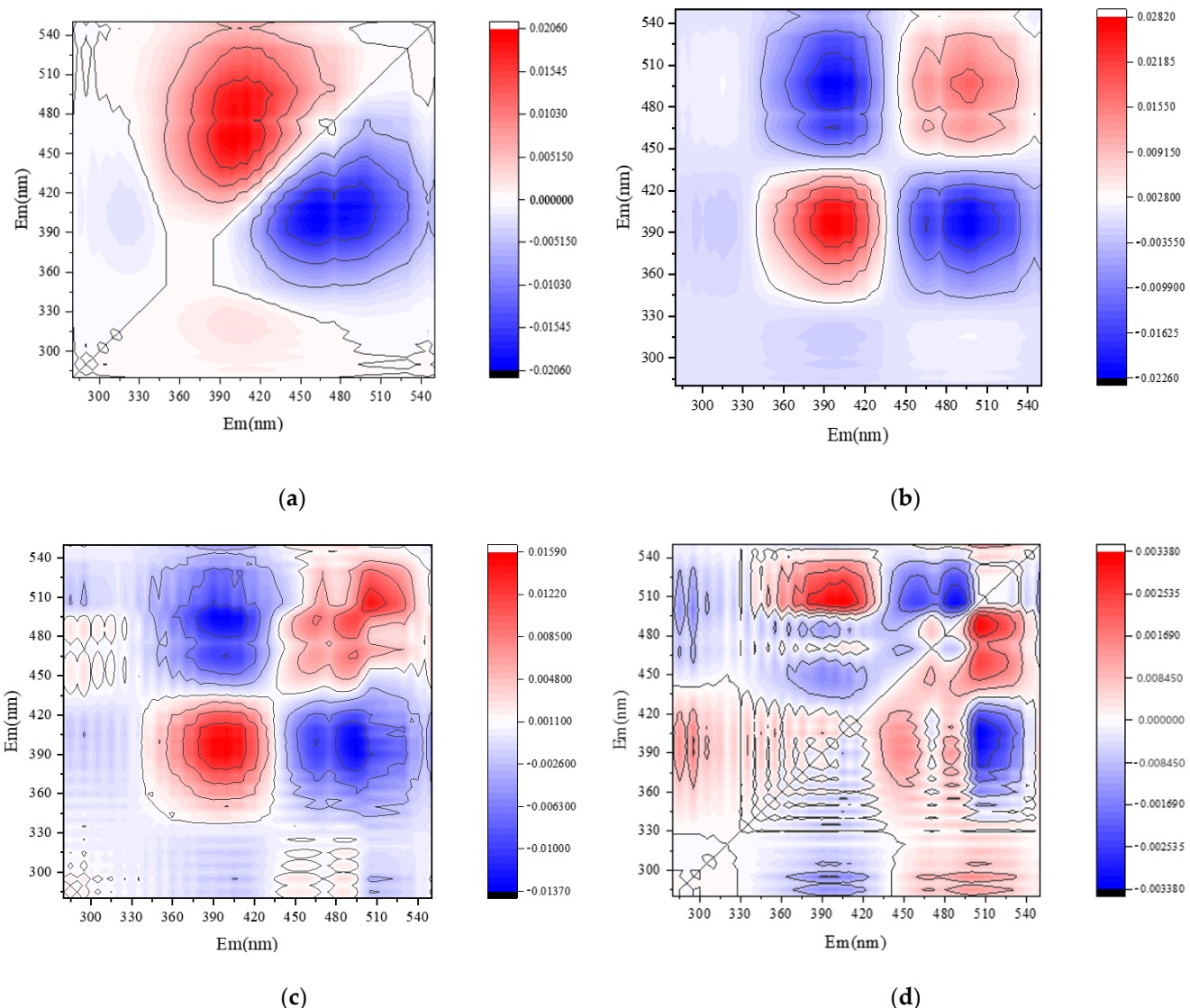

**Figure 6.** Two-dimensional correlation analysis of soil DOM fluorescence components. Note: CSCD and AI were conventional and aerated subsurface drip irrigation treatments, respectively. (**a**) Synchronous fluorescence spectra under CSDI. (**b**) Asynchronous fluorescence spectra under CSDI. (**c**) Synchronous fluorescence spectra under AI. (**d**) Asynchronous fluorescence spectra under AI.

**Table 6.** The change order table of DOM components with time as disturbance factor in CSDI.

| DOM Components | | C1 410 nm | C2 465 nm |
|---|---|---|---|
| C1 | 410 nm | + | − (+) |
| C2 | 465 nm | | + |

**Table 7.** The change order table of DOM components with time as disturbance factor in AI.

| DOM Components | | C1 410 nm | C2 465 nm | C3 450 nm |
|---|---|---|---|---|
| C1 | 410 nm | + | −(+) | −(+) |
| C2 | 465 nm | | + | +(−) |
| C3 | 450 nm | | | + |

By the combination of the synchronous (Figure 6a,c) and asynchronous (Figure 6b,d) plots, the effect of different irrigation treatments on the dynamic characteristics of soil DOM fractions can be analyzed on consideration of time as a perturbation factor. Under the CSDI treatment, and the changing order of DOM components under CSDI was 465 nm and 410 nm. When the symbols of C1 and C2 components in the synchronous graph (Figure 6a) and the asynchronous graph (Figure 6b) are the same, the change of spectral intensity at C1 wavelength is prior to that at C2 wavelength (Table 6). Similarly, the changing order under CSDI treatment was similar to that under AI treatment, the change of the fluorescence components under CSDI sorted in descending order was 450 nm, 465 nm, and 410 nm, indicating that the change order of DOM components with time was C3, C2, and C1 (Table 7).

### 3.4. Evolutionary Characteristics of Soil DOM Fluorescence Components Correlation Analysis of Soil DOM Fluorescence Components

Pearson's correlation analysis on soil DOM fluorescent substances was applied to clarify the correlation between various substances. Figure 7 showed that tyrosine-like protein was extremely positively correlated with tryptophan-like protein substance ($p < 0.01$) and negatively correlated with soluble microbial metabolism and humic acid ($p < 0.01$). Tryptophan-like protein was extremely negatively correlated with soluble microbial metabolites and humic acid ($p < 0.01$); ferulic acid was extremely positively correlated with soluble microbial metabolites and humic acid ($p < 0.01$). Soluble microbial metabolites were extremely positively correlated with humic acid ($p < 0.01$).

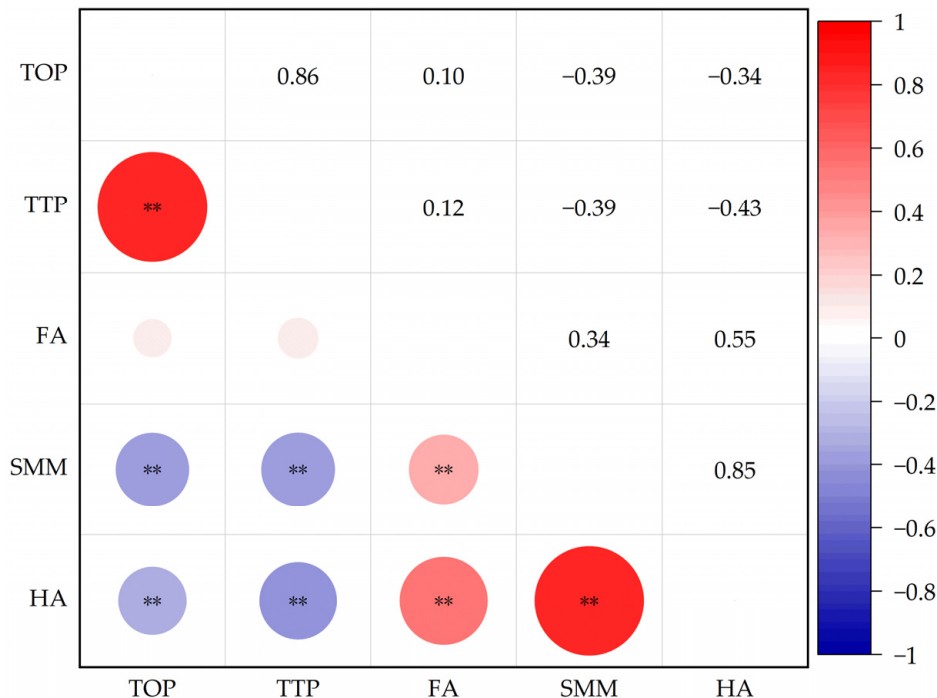

**Figure 7.** Correlation among soil DOM fluorescence components. Note: TOP means tyrosine-like proteins; TTP means tryptophan-like proteins; FA means fulvic acid; SMM means soluble microbial metabolites; HA means humic acid. The figure shows the correlation between the measured indicators. ** indicate a significant correlation at $p < 0.01$, respectively. The colors and numbers indicate the degree of correlation between the corresponding row comment and column comment, red is a positive correlation, and blue is a negative correlation.

## 4. Discussion

### 4.1. Effects of Aerated Irrigation on Soil DOM Components

Many studies in recent years have found that the mineralization of soil organic matter was higher than that of the aerobic strip under flooded conditions [33]. In this experiment, the response of humic acid-like substance to soil aeration was opposite under conditions of low nitrogen and low water and high nitrogen and high water. During the flowering and fruit-setting stage and the ripening stages, the relative contents of tyrosine-like protein substances and tryptophan-like proteins in the AI treatment were lower than those in the CSDI treatment under the conditions of low water with low nitrogen and low water with normal nitrogen. During the flowering and fruit-setting stage, fruit expansion stage, and ripening stage, the relative contents of fulvic acid-like substances, dissolved microbial metabolites, and humic acid-like substances were lower than those in the CSDI treatment. According to the findings, soil aeration promoted the loss of soil DOM components and significantly reduced DOM content in low water with low nitrogen and low water with normal nitrogen. On the one hand, soil aeration may affect the content of DOM by changing the soil environment [15]. Besides, the test soil was clay loam, which has high stability and low solubility of organic matter, and because of the double effect of high temperature in the shed from April to late July of planting time [23,34], which improved the soil adsorption capacity to DOM and accelerate nutrient turnover. Soil aeration could reduce the soil water saturation, thus, improving soil porosity [18], weakening the adsorption capacity to DOM, and helping in the release of soil DOM. Besides, it could enhance the contact of microorganisms and enzymes with DOM, thereby improving its biological effectiveness [22], increasing microbial activity, and allowing microorganisms to preferentially degrade DOM. According to related research, microorganisms can easily degrade the components in soil DOM ($\geq$85%) [35]. Soil aeration, on the other hand, has an impact on crop growth, and DOM is consumed by crops as a nutrient supplier.

Although DOM was the most active soil C pool in SOM, despite accounting for only a small part of soil organic matter (SOM) [36]. It not only supplied energy and material sources to indigenous microorganisms, but it was also a potential supplier of available nitrogen and phosphorus nutrients for crops [22]. Soil DOM could be directly absorbed and used by plants or decomposed to produce nutrients via low molecular weight. The active part could be directly absorbed and assimilated by plants to promote plant growth and development [37]. Soil aeration could increase the total surface area and active absorption area of crop roots. It could also significantly improve root activity [38], make root metabolism vigorous, and promote the absorption and utilization of soil DOM by the pepper to some extent. This might also be the potential reason for soil aeration to promote soil DOM component consumption.

However, when there is a lot of water and a lot of nitrogen, the results are different. Soil aeration could promote the consumption of small molecule protein-like substances and the accumulation of humus-like substances with a higher molecular weight under conditions of high water with low nitrogen and high water with normal nitrogen. The C3 component in the experiment is a typical humic acid-like component with a more complex structure and larger molecular weight than the C1 and C2 components [39]. The reason may be that aerated irrigation is beneficial to the formation of macromolecular DOM, while the environment with relatively sufficient water, nutrients, and soil aeration was more conducive to the growth of pepper and promoted pepper to absorb more soil water. It created relatively dry environmental conditions, resulting in small molecules in soil DOM being decomposed and consumed easily, since the drought could lead to the condensation and accumulation of humus. This was consistent with the findings of Wang et al. [28], who investigated the fluorescence characteristics of soil soluble organic in the loess hilly area matter under different vegetation. It was also possible that an environment with relatively good soil oxygen was more conducive to the release of soil DOM. Because soil microorganisms prefer DOM molecules with lower molecular weight for metabolism [40], the surplus content of small molecule, protein-like substances could satisfy microorganism

metabolism of microorganisms as well as crop absorption and utilization, which has slowed the consumption of substances with higher molecular weight to some extent. Microbes began to use aromatic ring substances [41] and were planted to preferentially absorb the lower molecular weight and stronger activity parts of soil DOM [35], when the easily degradable components decomposed to the low concentration.

In addition, under certain conditions, different components of soil organic matter could be synthesized and transformed, and part of protein-like substances could be transformed into more stable humus [42], potentially filling a gap in the consumption of humus substances. In this research, tyrosine-like proteins were positively correlated with tryptophan-like proteins ($p < 0.01$) and negatively correlated with humic acid-like substances ($p < 0.01$), and tryptophan-like proteins were negatively correlated with humic acid-like substances ($p < 0.01$).

Vinks et al. [43] found that flooding caused incomplete mineralization of soil organic matter, and the degradation products significantly increased DOM content. The alternation of dry and wet had the greatest impact on the relative content of protein-like components [44]. This experiment showed that increasing the amount of irrigation in the CSDI treatments, no matter at a low nitrogen level or normal nitrogen level, could significantly increase the content of tyrosine-like protein and tryptophan-like protein in three growth periods and significantly reduce the content of fulvic acid-like substance and humic acid-like substance in the late growth period of pepper. An increase in the irrigation rate can increase the content of DOM components and could be promoted in AI treatments. This study also discovered that the contents of tyrosine-like proteins, tryptophan-like proteins, fulvic acid-like substances, and humic acid-like substances increased first and then decreased during the pepper growth process, with the content being highest during the fruit expansion period. This could be because the pepper grew rapidly during the fruit expansion period. That may be due to the physiological metabolism of the root system being strong, resulting in high production of root exudates and root litter during the fruit expansion period. The high microbiological activity resulted in a relatively high nutrient turnover in the rhizosphere.

### 4.2. Effect of Aerated Irrigation on Soil DOM Spectral Index

To characterize soil DOM under different irrigation modes, the fluorescence spectral indices FI, BIX, and HIX were used. FI was defined as the ratio of emission wavelengths at 470 nm to that at 520 nm when an excitation wavelength was 370 nm [45], which could be used to distinguish the source of DOM [46]. When FI $\leq$ 1.40, the soil water-soluble organic matter primarily came from plants, with a low autogenous source input. When FI $\geq$ 1.90 was discovered, it was primarily due to the metabolic activities of microorganisms with obvious autogenesis characteristics of autogenesis. The combination of plants and microorganisms produced 1.40 < FI < 1.90. Under CSDI and AI, the fluctuation range of soil DOM fluorescence index during the entire growth period was 1.63–1.76 and 1.66–1.81, respectively, between 1.40 and 1.90. It indicated that the soil DOM came from the mixture of plants and microorganisms under the two irrigation modes. The intensity of recent biological activities, an index that measured the degree of autochthonous pollution, could be associated with BIX [47].

Autogenous contribution to soil DOM was relatively low when 0.60 < BIX < 0.80. There were many new autogenous components in soil DOM components when 0.80 < BIX < 1.00. When BIX > 1.00, the DOM in the soil was primarily neotenic. In this study, the fluctuation range of the biological index under CSDI and AI was 0.67–0.72 and 0.65–0.72, respectively, indicating that under both CSDI and AI irrigation modes, the proportion of autogenous organic matter in soil DOM components was relatively low. Some studies have also shown that both exogenous input and endogenous production of DOM released from sediments occurred as dissolved oxygen levels rise [47]. The BIX value of DOM in the sediment ranged from 0.6 to 0.8 at a dissolved oxygen level between 0% and 25%, and as the oxygen content increased, the released DOM components contained more new autogenesis. In this study,

the new autogenous sources of DOM components in soil under AI were not obvious. We hypothesized that the autogenic source index is low; microorganism availability is reduced; the components representing DOM tend to be stable, making them difficult to degrade by microorganisms; and that the relatively low level of dissolved oxygen in irrigation water was the cause.

Besides, it is also due to complex factors such as microbial degradation, chemical polymerization, and pepper absorption. When the FI and BIX index were combined, it was discovered that soil DOM in both irrigation modes had the same source as both internal and external inputs, with the external inputs being the main source and the internal inputs being supplementary HIX could represent the humification of DOM. The greater the degree of humification, the higher the value [45]. The primary sources of DOM were plant debris, soil humus, microorganisms, and rhizosphere secretions. The decomposition of plant residues produced transitional organic matter, which contributed significantly to the accumulation of soil organic carbon [48]. The primary form of organic matter for surface soil was DOM, and it was an important index for assessing soil fertility [49]. The humification degree of soil DOM was weaker when HIX < 4, and soil DOM exhibited obvious humification characteristics when 10 < HIX < 16 [15]. The HIX value of AI treatment at the maturity stage of the pepper ranged from 8.47 to 9.94 in this research. The humification degree of AI increased by 57.11% and 78.36% compared with that under the conditions of a high amount of irrigation and both low nitrogen and normal nitrogen ($p < 0.05$). The humification degree of high water and low nitrogen and high water and normal nitrogen treatment decreased by 49.58% and 40.62%, respectively, compared with that of low water and normal nitrogen treatments ($p < 0.05$). This could be attributed to relatively high irrigation amount under high water conditions and high irrigation frequency in the greenhouse from April to mid-July, resulting in the breakdown of large soil aggregates, the loss of soil organic carbon and nitrogen mineralization and the aggravation of DOM leaching loss [50].

To summarize, soil aeration was a practice that combined land use and nutrification, which could weakly reduce the carbon and nitrogen sink functions of soil and was beneficial to the maintenance and development of the ecological micro-environment of facility vegetable. This study, however, only discussed soil DOM components, spectral index, and humification degree. To fully understand the mechanism of AI on land nutrification, it was necessary to broaden the aeration gradient and long-term positioning test and further research the related microorganisms and enzymes.

## 5. Conclusions

(1)　The composition of soil DOM under aerated irrigation showed that soil DOM components are dominated by humic acid-like substances, fulvic acid-like substances, tryptophan-like proteins, and supplemented by tyrosine-like proteins and dissolved microbial metabolites. Soil aeration could promote the consumption of soil DOM components under low irrigation. In contrast, soil aeration can accelerate the depletion of small molecular proteins and the accumulation of humus substances with higher molecular weight under high irrigation amounts. Plants can make better use of soil mineral nutrients under aerated irrigation. At the mature stage of pepper, the humification index of AI treatments ranged from 8.47 to 9.94 during the maturity growth stage of pepper, averagely increased by 31.59% compared with that in conventional subsurface drip irrigation.

(2)　Aerated irrigation is beneficial to the maintenance of soil fertility, has a positive effect on the DOM properties of the soil. To promote aerated irrigation in various environments, it was necessary to further study the influence mechanism of aerated irrigation on DOM humification process in long-term field conditions.

**Author Contributions:** Conceptualization, R.X. and H.L.; investigation, Z.X., G.Y., K.S. and Y.D.; methodology, Y.Z. and H.P.; project administration, H.L.; software, Z.X.; supervision, Y.Z.; validation, Y.Z., H.P. and Y.H.; visualization, Z.X., G.Y. and J.Y.; writing—original draft, R.X.; writing—review and editing, R.X. and H.L. All authors have read and agreed to the published version of the manuscript.

**Funding:** This research was funded by the National Natural Science Foundation of China (No. 52079052), Key Science and Technology Project of Henan Province (No. 212102110032), the Key Research and Development Program Major Science and Technology Innovation Project in Shandong Province (No. 2019JZZY010710).

**Institutional Review Board Statement:** Not applicable.

**Informed Consent Statement:** Not applicable.

**Data Availability Statement:** Data are contained within the article.

**Acknowledgments:** We fully appreciate the editors and all anonymous reviewers for their constructive comments on this manuscript.

**Conflicts of Interest:** The authors declare no conflict of interest. The funders had no role in the design of the study; in the collection, analyses, or interpretation of data; in the writing of the manuscript; or in the decision to publish the results.

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
