# Peer review of "The Influence of Aerated Irrigation on the Evolution of Dissolved Organic Matter Based on Three-Dimensional Fluorescence Spectrum"

_agronomy, doi:10.3390/agronomy13040980_

Round 1

Reviewer 1 Report

Some correction suggestions are shown on the manuscript. Examples:

Introduction section:

-In the introduction, a few sentences can be given about aerated irrigation. which irrigation method is applied etc. way...

experimental site

-relative humidity looks very fluctuating. It can be explained about.

trial management

-what type of pepper? pointed, bell, green or red

-In what medium were the seedlings grown?
-It would be nice to add a small figure or picture about the venturi.

Reviewer 2 Report

The work is interesting, has a clear degree of originality, and is appropriate for publication in the journal after performing a major and very careful revision. Nevertheless, it needs some further improvements. In general, there are still some occasional grammar errors throughout the manuscript, especially the article "the," "a," and "an" are missing in many places; please make spellchecking in addition to these minor issues. The reviewer has listed some specific comments that might help the authors further enhance the manuscript's quality.

The Abstract section is not giving any information about methodology and recommendations, and especially about the importance of the subject as it should be with clear. I suggest the authors to remove generic lines and present the strong statements and novelty of article. The abstract written by qualitative sentences. It is need to modify and rewrite based on the most important quantity results from this research. The abstract should be redesigned. You should avoid using acronyms in the abstract and insert the work's main conclusion.

The objectives should be more explicitly stated. It is better to improve your contributions which are not so clear to show the advantage of your work. The Introduction section must be written on more quality way. The research gap should be delivered on more clear way with directed necessity for the conducted research work. What is the novelty of this work? The novelty of this work must be clearly addressed and discussed in Introduction section. In the last sentence of the introduction, a hypothesis should be addressed and the objectives of the study should be stated. Please elaborate on the introduction section. The following literature may be helpful in this regard:

-Improvement of Water and Crop Productivity of Silage Maize by Irrigation with Different Levels of Recycled Wastewater under Conventional and Zero Tillage Conditions, Agricultural Water Management, 277, 108100

-CO2 Emission From Soil in Silage Maize Irrigated with Wastewater under Deficit Irrigation in Direct Sowing Practice, Agricultural Water Management, 271, 107779

-Soil CO2 Emission Linearly Increases with Organic Matter Added Using Stabilized Sewage Sludge under Recycled Wastewater Irrigation Conditions, Water Air and Soil Pollution, 234 (59), 1-18

-Infuences of Farmyard Manure and its Biochars Prepared at Diferent Temperatures on Soil Properties and Soil Enzymes and CO2 Emission from Soil Under İrrigation Conditions with Treated Wastewater, Water Air and Soil Pollution, 234 (56), 1-22

-CO2 emissions and their changes with H2O emissions, soil moisture, and temperature during the wetting–drying process of the soil mixed with different biochar materials, Journal of Water Climate Change, 13 (12), 4273-4284.

-Deficit Irrigation with Wastewater in Direct Sowed Silage Maize Reduces CO2 Emissions from Soil by Providing Carbon Savings, Journal of Water Climate Change, 13 (7), 2837-2846.

-Quality Proficiency to Crop, Soil and Irrigation System of Recycled Wastewater from the Van/Edremit Wastewater Treatment Plant, Yuzuncu Yıl University Journal of Agricultural Sciences, 32 (3), 497-506.

-CO2 Emission from Soil Containing Different Organic Manures in Wetting-Drying Conditions and the Relationships of CO2 Emission with Moisture, Temperature and H2O Emission, Journal of Agricultural Faculty of GaziosmanpaÅŸa University, 39 (3), 161-168.

Some soil properties are presented in the material method section. However, while the N, P and K contents of 0-20 cm depth are given, the bulk density, pH and field capacity properties of 0-40 cm depth are given. All these properties must be given for both soil layers. It can even be presented as a table if possible. In addition, which analysis approaches were used to determine these features? The last sentence in which soil texture is given should be written as follows; the particle size distribution made according to the …….., it was determined that the soil with ………. texture (Sand: 33%, Silt: 34% and Clay: 33%). In the material and method section, the definition of irrigation practices can be examined under a separate heading. Are the parcel sizes appropriate? Isn't it too small? Why was the coefficient of the evaporation vessel taken as 0.6 and 1.0. The irrigation part needs more detail. Why 0.05 probabilities were used in the statistical analysis part. Are the correlations in Figure 6 obtained with RStudio? Or another program. This should be stated in the statistical analysis section.

Results; This section is well written. Discussion; Overall, the discussion part is weak. The Discussion should summarize the manuscript's main finding(s) in the context of the broader scientific literature and address any study limitations or results that conflict with other published work. Conclusion; This part should be written from the beginning. In addition, itemization should not be done and statistical findings should be avoided, and perhaps the most important; some future works should be added to your conclusion because no suggestions were made. Please elaborate it a bit more. Visual readability of Figures 1 and 2 is rather poor. Is pixel optimization possible? Many literature the couple of references mentioned. Kindly be specific and recheck its availability. There are lots of commas and full stops are missing.

Round 2

Reviewer 2 Report

I think some of the revisions mentioned earlier were overlooked. Please make a more careful revision. Also, answer the suggestions more carefully. best regards.

Round 3

Reviewer 2 Report

Dear author,

The authors have fully made all the corrections mentioned. 

Author Response

Thank you for your suggestions for changes to the manuscript.